# Rapid Detection and Elimination of Subsurface Mechanical Damage for Improving Laser-Induced Damage Performance of Fused Silica

**Qingzhi Li, Yubin Zhang \*⬤, Ting Shao, Zhaohua Shi, Jin Huang, Xin Ye \*⬤, Liming Yang and Wanguo Zheng**

Research Center of Laser Fusion, China Academy of Engineering Physics, Mianyang 621900, China; dearlqz@163.com (Q.L.); shaot05@163.com (T.S.); jinhuangcaep@163.com (J.H.); yangliming621000@126.com (L.Y.); wgzheng_caep@sina.com (W.Z.)
* Correspondence: zhangyub@mail.ustc.edu.cn (Y.Z.); xyecaep@mail.ustc.edu.cn (X.Y.);
   Tel./Fax: +86-0816-2480872 (Y.Z. & X.Y.)

**Abstract:** The fabrication of SSD-free fused silica optics is a crucial objective for high-power laser applications. To treat the surface of polished fused silica, a combination of RIE/RIBE and deep-controlled etch (DCE) techniques are typically employed. Currently, it is important to consider and study the ideal etching depth and precision while using combined etching techniques to remove the identified SSD. Herein, we present a novel approach to identify the distribution of SSD in fused silica, which corresponds to a specific grinding/polishing process condition. Our method involves using a mobile RIBE to perform cone cutting and remove material from the polished fused silica surface. Afterward, we etch the optical element's surface with HF to visualize the subsurface cracks and understand their relationship with the RIBE depth. Through a systematic investigation of the combined etching technique, we establish a correlation between the depth of RIBE and DCE and the performance of laser damage. The combined etching technique can be implemented as a dependable approach to treat the surface/subsurface defects in fused silica and has the potential to improve laser damage resistance significantly.

**Keywords:** fracture; optical microscopy; lasers; silica





## 1. Introduction

The presence of subsurface damage (SSD) on the final fabricated fused silica optic was found to negatively affect its mechanical strength and laser damage performance [1,2]. In high-power laser facilities, such as the National Ignition Facility (NIF), Laser Megajoule (LMJ), and Shenguang-III, these effects are particularly pronounced and extensive. Therefore, the detection and elimination of the SSD layer on the fused silica optical surface can lead to significant practical benefits that enable the advancement of optical fabrication processes [3–5].

Over the last few decades, researchers have conducted extensive investigations on subsurface damage (SSD), focusing on various questions related to its development and techniques for mitigating or eradicating it [6–8]. Brittle-fused silica material typically experiences SSD due to grinding and polishing processes [9,10]. Surface cracks are generally identified by macroscopic evaluations as scratches and digs in the subsurface layer and serve as reservoirs absorbing precursors that heat up and explode upon exposure to high-fluence laser light, particularly during nanosecond pulses at 351 nm [11–15]. To fabricate SSD-free fused silica optics, researchers have developed a range of posttreatment techniques, including magnetorheological fluid finishing (MRF), ion beam etching (IBE), reactive ion/ion-beam etching (RIE/RIBE), and HF-based wet etching [16,17]. Among the available techniques, the HF-based etching route (which typically involves optimized and deionized water cleaning and the HF/NH4F etching process under ultrasonic/megasonic

conditions described as AMP or DCE previously [18]) has demonstrated high effectiveness in exposing SSD and increasing the laser damage resistance of fused silica optics. However, this wet etching process leaves traces of SSD, leading to an increase in surface roughness and errors and resulting in a non-linear effect when the laser irradiates the optical element, causing a reduction in the damage threshold.

An alternative approach proposed to eliminate the sub-surface mechanical damage of fused silica is to combine RIE/RIBE and DCE, as described in detail elsewhere [19]. This method represents a significant advancement in the production of high-quality silica-fused optics because it results in traceless etching patterns and prevents the introduction of additional defects, such as redeposited silica compounds. Consequently, this process creates an SSD-free fused silica optical surface with an enhanced laser-induced damage threshold and surface quality compared to traditional HF-based deep etching. To maximize the damage threshold, the matching relationship between the RIE/RIBE depth and DCE depth is critical during the combined treatment of fused silica. In our previous studies, the commonly applied RIE and DCE depths were 1 μm and 3 μm, respectively, resulting in a 2.4–2.7 times increase in the zero-probability damage threshold relative to the original polished surface [2,18,20,21]. We extended the previous work by combining 1.5 μm RIBE and 3 μm DCE, resulting in a 2.4 times increase in the damage threshold [22]. Other results demonstrate that higher RIE and DCE depths, specifically 5 μm and 3 μm, are required to increase the zero-probability damage threshold by at least 2.7 times [18]. The difference in the optimal matching relationship between RIE/RIBE depth and DCE depth is attributed to differences in the finished operation of the optic, including the particle size of the abrasive used during the grinding process and polishing type.

For fused silica surfaces that have been ground and polished, the question remains whether there exists a practical method to determine the optimal depths of RIE/RIBE and DCE. Additionally, the accuracy and confidence of the method may become problematic when utilizing the combined etching technique to remove the detected SSD layer and maximize the laser-induced damage threshold of the optic. Initially, we utilized mobile RIBE to remove tapered material from the previously polished and ground silica surface. We then etched the optical surface using HF acid to enable the visualization of the subsurface fractures as a function of RIBE depth. We determined the SSD distribution of the optics by examining the fracture features uncovered through the use of HF acid. In order to verify its soundness, we systematically investigated the relationship between RIBE/DCE depth and laser damage performance through the utilization of a combined etching processing technique. The obtained results confirm the feasibility and accuracy of this methodology.

## 2. Materials and Methods

### 2.1. Materials

Fused silica specimens measuring 50 mm × 50 mm × 10 mm were provided by the Chengdu Fine Optical Engineering Research Center in Chengdu, China. The optical surfaces of these specimens were ground and polished using a consistent process. This process has been continually refined to establish a standard technical route.

### 2.2. SSD Detecting Experiment

A surface profile of the ion beam spot was characterized using white light interferometry (NewView 8300 Surface Profiler, Zygo, Middlefield, CT, USA).

Figure 1 presents a schematic of the process used to detect the sub-surface damage (SSD) on a polished fused silica surface. The first step involves polishing and grinding the fused silica samples in a traditional manner. A technique known as reactive ion beam etching (RIBE) is then utilized to remove a predetermined thickness of the sample's surface in a gradient section [23,24]. This process involves scanning the optical surface line-by-line with a well-characterized RIBE removal function, gradually increasing or decreasing the scanning velocity. Before commencing the etching process, it is important to address whether

the scanning velocity has a linear effect on the etching rate as the thermal distribution on the sample's surface changes with velocity.

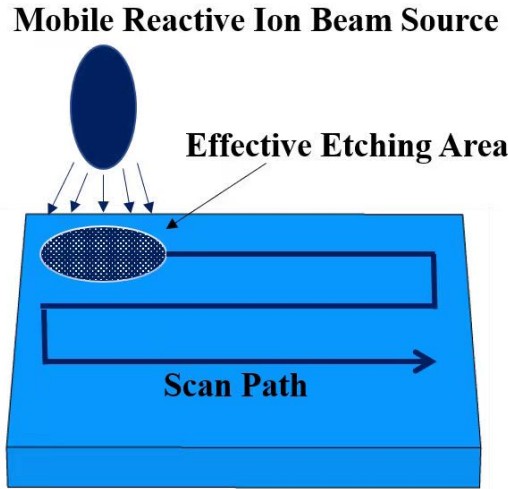

**Figure 1.** Schematic diagram of the RIBE etching process.

We provide a schematic diagram of the ion beam etching principle in Figure 1. In our past research, we found that the etching rate was linearly dependent on the scanning velocity, indicating that the thermal effect induced by the ion beam has a weak impact on the material removal characteristic [25]. Therefore, we accomplished removal depths ranging from 100 nm to 1 μm on the entire sample surface, covering an area of 50 mm × 50 mm.

Wet etching, which employs an aqueous acid or base solution, is a widely used method to prepare the samples for subsequent processing and characterization [26]. Specifically, HF-based etching is suitable for reliable opening fractures that are present at the optical surface but are closed or optically contacted to neighboring material, which can be difficult to observe during a microscopic examination. This type of etching is also effective in revealing subsurface damage that is generated during the fabrication process and has been subsequently buried under a re-deposited layer of refractive index-matched hydrated glass. A 15 min etch using a 1:4:10 $HF/NH_4F/H_2O$ solution allows for adequate surface development by opening fractures with the removal of about 4 μm of material from the optic surface. We give a schematic representation of the steps of the gradient detection method in Figure 2.

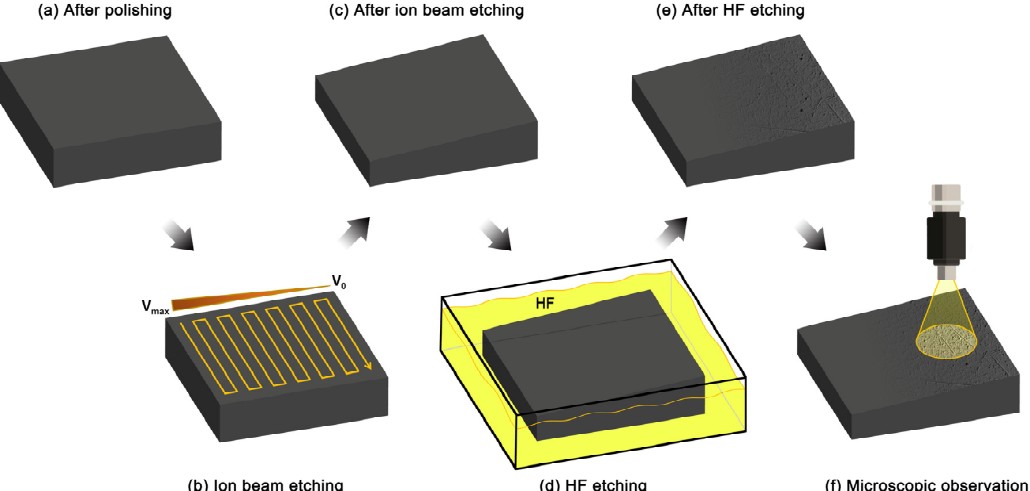

**Figure 2.** Schematic illustrating the step in the gradient detecting technique to determine the SSD depth distribution. (**a**) Traditional finishing operation (grinding and polishing), (**b**) reactive ion beam

etching to create a precise, shallow, linear wedge (over a 50 mm square patch) with removal depths ranging from 100 nm to 1 μm, (**c**) dynamic chemical etching (multi-frequency agitation; 49% HF:30% NH$_4$F:H$_2$O ratio of 1:4:10) for 15 min to expose the cracks on the surface, and (**d**) analyzed using bright field transmission optical microscopy at 50× and 500× magnifications at various distances along the surface wedge, (**e**) after HF etching, the samples were rinsed with deionized water and then dried naturally, (**f**) analysis of surface etching using optical microscopy.

### 2.3. Combining Etching Experiment

A description of the removal amount chosen for the RIBE and DCE experiments of the samples is shown in Table 1.

**Table 1.** Summary of amount removed by RIBE and DCE for polished fused silica samples.

|  | Sample | RIBE Depth (nm) | DCE Depth (μm) |
|---|---|---|---|
| 1st experiment | A1 | No etch | 10 |
|  | A2 | 200 | 3 |
|  | A3 | 400 | 3 |
|  | A4 | 600 | 3 |
|  | A5 | 800 | 3 |
|  | A6 | 1000 | 3 |
| 2nd experiment | B1 | No etch | 10 |
|  | B2 | 400 | 0.5 |
|  | B3 | 400 | 1 |
|  | B4 | 400 | 3 |
| 3rd experiment | C1 | No etch | 10 |
|  | C2 | 400 | 0.5 |
|  | C3 | 400 | 1 |
|  | C4 | 400 | 3 |
|  | C5 | 400 | 6 |

## 3. Results and Discussion

### 3.1. SSD Detecting Experiment

We created a nanostructure using reactive ion beam etching (RIBE) with etching depths ranging from 0 nm to 1000 nm. By observing the depth of the etched defects, we could indirectly determine the depth of the single-crystal region. We guided the RIBE depth to the minimum depth while maintaining an etching depth of 0 nm, allowing us to observe the depth of the single-crystal region through an optical microscope.

We used RIBE to create an inclined surface of 50 mm × 50 mm components with different etch removal rates. The inclined surface denotes the region where the etch removal amount ranges from 0 to 1 micron. The inclined surface denotes the region where the etch removal amount ranges from 0 to 1 micron. An indirect estimation of the subsurface damage layer depth is achievable through this technique. Then, the subsequent guidance for RIBE removal depth is obtainable. Eventually, minimizing the etch removal amount becomes feasible. An examination of the optical microscope image is shown in Figure 3, where the damaged layer under the surface is absent at an etch removal of 300 nm.

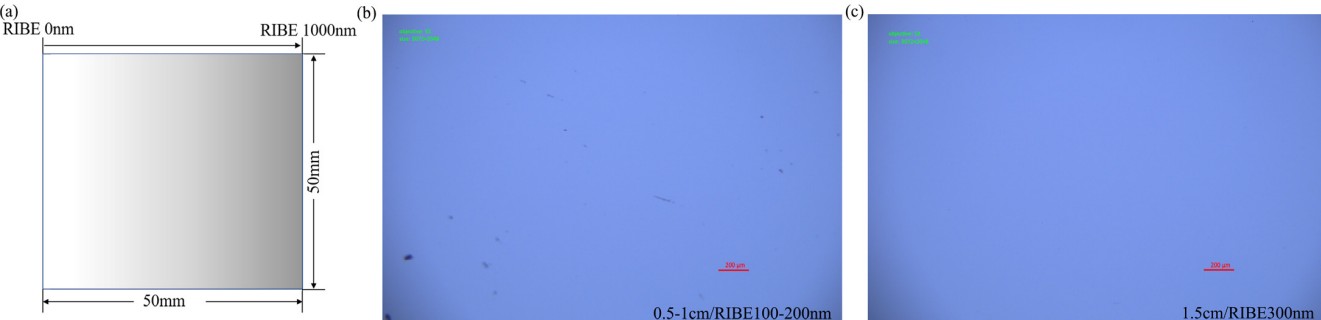

**Figure 3.** Ramp etching principle and shape. (**a**) A schematic diagram of the ramp etching process; (**b**) the surface profile of an optical element etched at 100–200 nm and 6–10 mm of the ramp element; and (**c**) the surface profile of an optical element etched at 300 nm at 15 mm of the ramp element.

### *3.2. Combining Etching Experiment*

(1)   RIBE depth and damage performance

We conducted an experimental study aimed at reducing the depth of the RIBE. Our experimental design included five different RIBE depths (200 nm, 400 nm, 600 nm, 800 nm, and 1 μm) that corresponded to a 3 μm DCE. As a reference point, a sample treated with a 10 μm deep DCE was utilized. Using a small-area laser system and a one-on-one strategy in compliance with 21254-1:2011 [27], we evaluated the damage performance of the etched samples [28]. For each type of laser energy, we randomly selected 20 sites on the exit surface of the samples. The testing system we used generated a near-Gaussian laser pulse of 5 ns at 351 nm, with a maximum output energy of approximately 18 J and a beam diameter of approximately 0.7 mm. The repetition rate of this system is one shot per minute. By analyzing the beam profile and measuring the absolute energy, we were able to determine *F*, the energy density locally at the exit surface of the samples, as follows:

$$F = \frac{E \times C \times K}{\pi \left(\frac{d}{2}\right)^2} \times f \tag{1}$$

where *E* is the absolute energy obtained from the meter [29]. *C* and *K* are the meter correction coefficient and spectroscopic coefficient, respectively. *f* is the modulation degree of the local energy in the laser beam profile. In Figure 4, it is observed that samples treated with 10 μm of DCE had the highest laser damage probability, as indicated by the dotted black fitting line. On the other hand, the sample treated with 200 nm of RIBE (see red circles) showed an evident increase in damage resistance where the damage probability never exceeded that of the sample treated with 10 μm DCE for the same fluence. Nonetheless, the results exhibit great fluctuations with the increment in laser fluence, suggesting that the etched surface is highly inhomogeneous. Also, the data imply that eliminating the SSD layer of the as-polished sample through a 200 nm RIBE treatment was challenging. Increasing the RIBE depth to 400 nm, 600 nm, or 800 nm led to a further improvement in damage resistance. Precisely, the damage probability was below 20% for all laser fluences below ~60 J/cm$^2$, as denoted by the orange dotted line. However, the damage resistance decreased significantly at ~70 J/cm$^2$ for all three samples, and the differences between the samples were insignificant, suggesting that the SSD layer depth ranged from 200 nm to 400 nm. Conversely, increasing the removal amount to 1 μm led to a decrease in damage resistance, although it was still superior to the 10 μm DCE-treated sample. In conclusion, the results emphasize the necessity of a 400 nm RIBE pretreatment during the combined etching of fused silica optics with fine polishing.

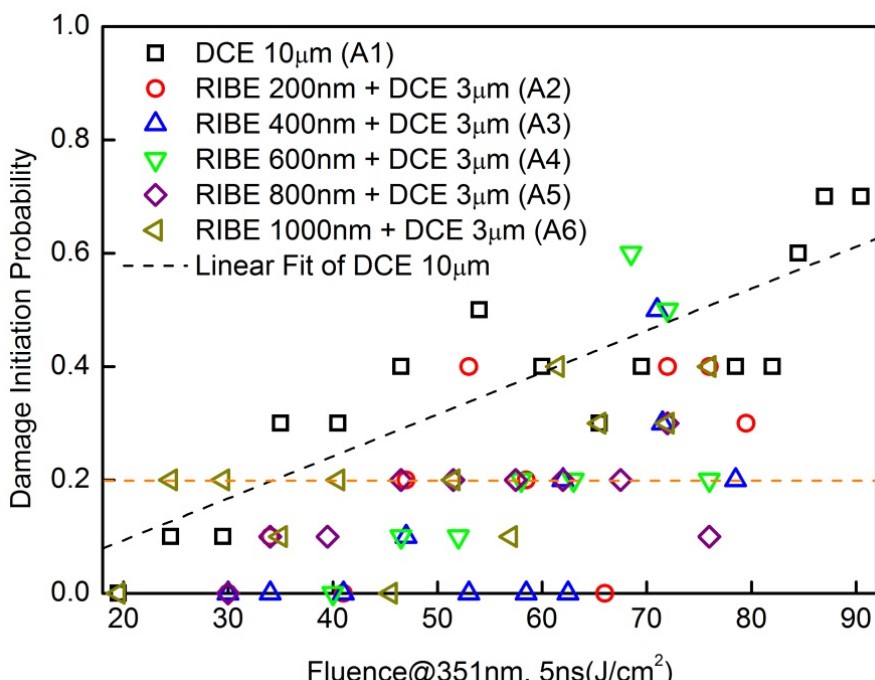

**Figure 4.** Small beam laser initiation damage probability as a function of the RIBE depth for the combined-etched samples (including a 10 μm DCE sample) is described in Table 1. The dotted black line represents the linear fitting result for the 10 μm DCE sample. The orange dotted line represents the upper bound (20%) of the laser damage probability for most of the combined processes at a laser fluence below ~60 J/cm$^2$.

(2) RIBE depth and surface morphology

Figure 5 displays the optical micrographs of the six samples that underwent combined etching with different RIBE depths or treatments with 10 μm of DCE. Figure 6 shows the surface morphology of the etched samples, which were examined using a white light interferometer. The test process was to use the white light interference principle, record the position information in a certain area of the sample surface, count, and obtain the surface roughness value of the optical element. The micrographs depict the change in surface morphology. Scratches and pits embedded in the subsurface layer of fused silica became visible after a single treatment with DCE. The increased amount of material removed enlarged the size of the exposed defects. In the case of the combined-etched samples, the RIBE pre-treatment could eliminate the subsurface damage (SSD) of fused silica without a trace, resulting in the almost complete absence of scratches or pits on their surfaces. Upon exposure to a 351 nm laser, there was a significant decrease in damage probability on the surfaces, particularly for the sample pretreated with 400 nm of material removal, where there was almost no damage initiated when the fluence increased from ~30 J/cm$^2$ to ~62 J/cm$^2$ (refer to Figure 4).

(3) DCE depth and damage performance

An experiment was conducted to investigate the effect of DCE depth on the damage performance of fused silica samples. The samples were pretreated with material removal of 400 nm. This study selected three DCE depths of 500 nm, 1 μm, and 3 μm. The parameters for the damage test were the same as those in the first experiment. Figure 7 displays the initiation of damage probability as a function of the DCE depth for all combined-etched samples, including a 10 μm DCE sample as a reference. The optimized depth obtained using RIBE indicates that 3 μm of material needs to be etched by HF acid to maximize the laser damage resistance of the fused silica samples. After the etching process, the zero-probability damage threshold increased up to approximately 75 J/cm$^2$. Although a

0.5 µm DCE retreatment seemed to have the same effect, a 1 µm DCE retreatment showed an increase in the damage probability, indicating that an insufficient DCE depth could not eliminate RIBE-introduced defects that cause high and/or volatile damage probability. Our previous studies [4] reported the fundamental characteristics of ion/ion-beam etching, which may be involved in this mechanism. Since the defects on the surfaces of the samples had a low density and uneven distribution, the features could be missed by standard small-area testing. Therefore, we recommend designing and conducting a large-area damage experiment.

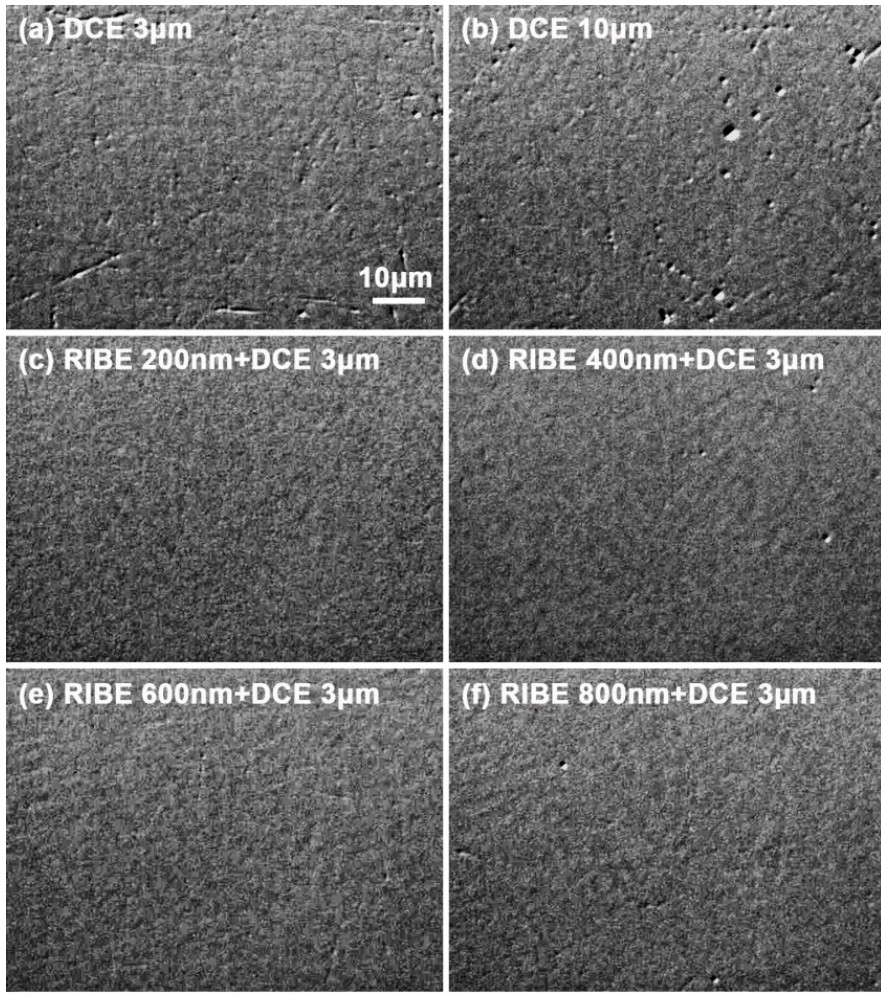

**Figure 5.** Optical micrographs of samples single-DCE-treated with 3 µm and 10 µm material removal (**a**,**b**) and combined-etched samples RIBE-pretreated with 200 nm, 400 nm, 600 nm, and 800 nm of material removal (**c**–**f**). The scales in (**b**–**f**) are all consistent with (**a**).

In the third round of experiments, samples with the same four etching parameters as the previous round were included. A deeper DCE treatment depth of 6 µm was added to observe its effect on the component damage threshold. Using the UV laser damage platform of Department 7 (the wavelength, pulse width, and laser repetition rate were 355 nm, 9.3 ns (FWHM), and 1 Hz, respectively), the experimental study was conducted using the "raster-scan" damage testing strategy. The continuous scanning of a square area of 30 mm × 30 mm on the sample surface was carried out using the same laser fluence that corresponded to 90% of the maximum laser fluence of the entire laser. This resulted in 30 × 30 test points over the scanning area. To increase the sampling area, the laser spot diameter was consequently expanded from 0.7 mm to 1.4 mm, resulting in an overlap of

spots between two consecutive laser shots. The "scan path and spot overlap model" used in this raster-scan damage testing strategy is shown in Figure 8.

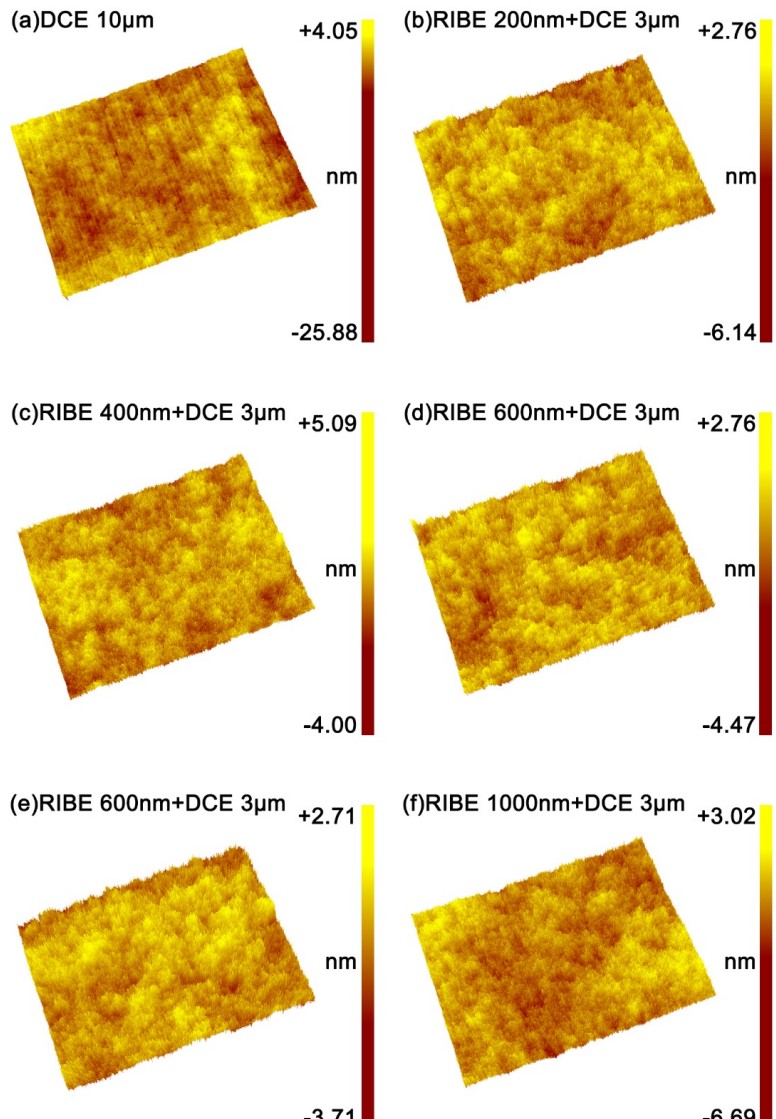

**Figure 6.** Corresponding RMS roughness values for all the samples. Surface roughness of samples single-DCE-treated with 10 μm of material removal (**a**) and combined-etched samples RIBE-pretreated with 200 nm, 400 nm, 600 nm, 800 nm, and 1 μm of material removal (**b–f**).

Finally, the number of damage sites under 900 laser shots was calculated for each sample, yielding the damage density shown in Figure 9. Testing revealed that combining 400 nm of RIBE with 3 μm of DCE produced an optimal increase in the fused quartz surface damage resistance. Only five damage sites appeared over the entire 900 mm$^2$ test area, which is a decrease of nearly one order of magnitude compared to samples treated with 10 μm DCE. This result is consistent with that of the second round of experiments. For the CEP samples treated with DCE depths of 0.5 μm and 1 μm, 30 and 27 damage sites were observed, respectively, indicating that insufficient DCE depth could not effectively remove the chemical structural defects on the surface of the component secondary to the RIBE process. This result also confirms the inference made in the second round of experiments, namely that an increase in the number of sample points is necessary to clearly distinguish the differences in damage performance between the two joint etching parameters. A deeper DCE treatment, such as 6 μm of DCE depth, did not achieve further enhancement in fused

quartz component damage resistance, suggesting that 3 μm of DCE treatment is sufficient to remove the secondary defect layer generated by RIBE processing while also indicating the existence of an optimal DCE removal amount for achieving the maximum increase in the damage resistance of fused quartz components treated with CEP.

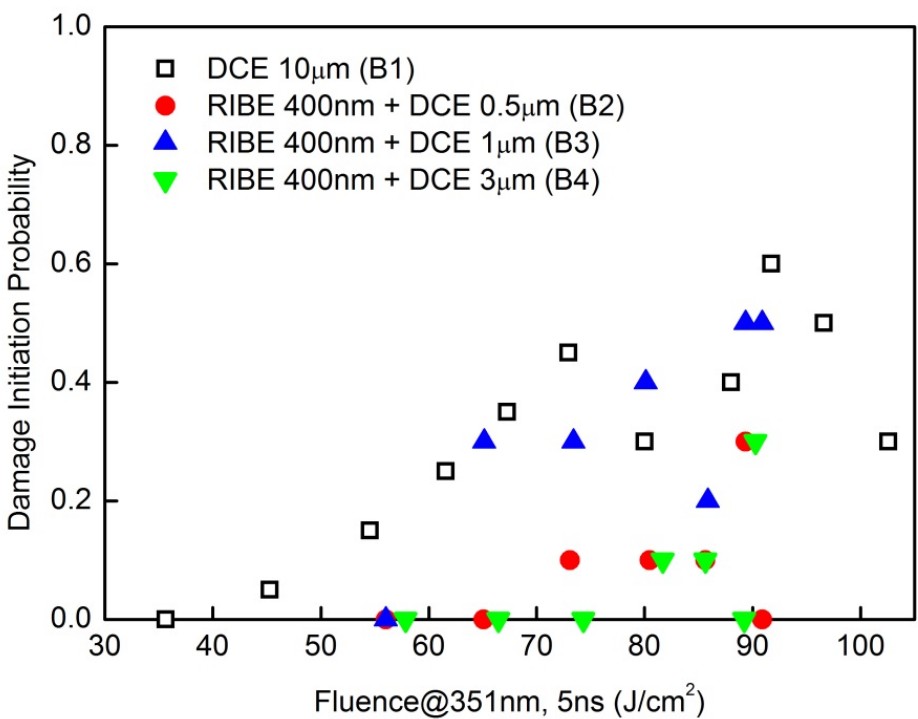

**Figure 7.** Small beam laser initiation damage probability as a function of DCE depth for the combined-etched samples (including a single DCE sample) is described in Table 1.

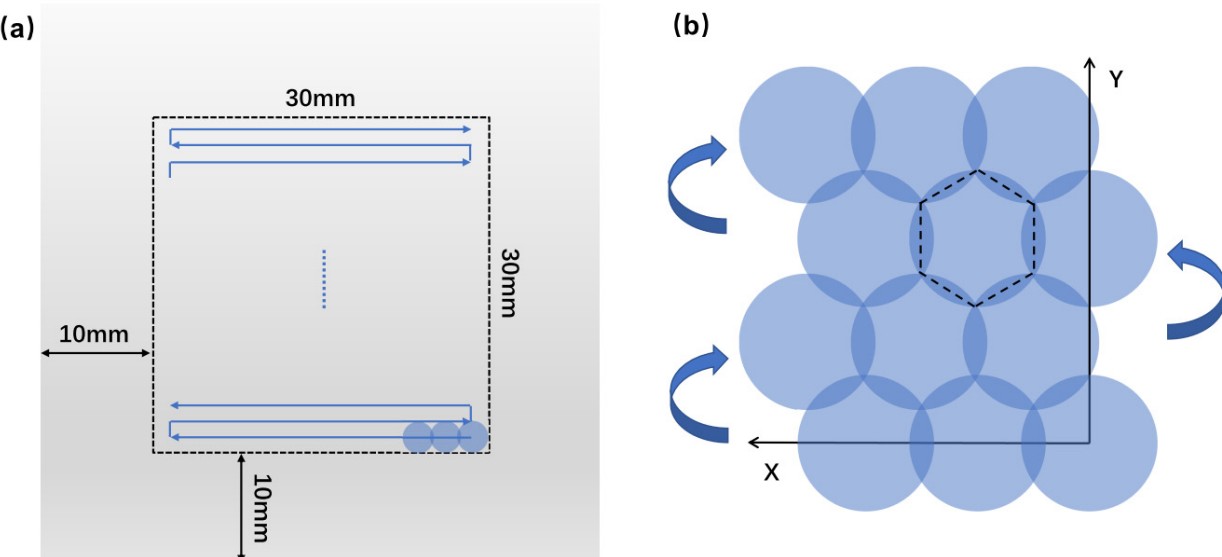

**Figure 8.** (**a**) Raster-scan path and (**b**) overlap model for the large area damage test, which corresponded to a total irradiated area of ~9 cm². In total, 900 sites (30 × 30) were successively irradiated in this region with a beam diameter of ~1.4 mm at ~11 J output energy.

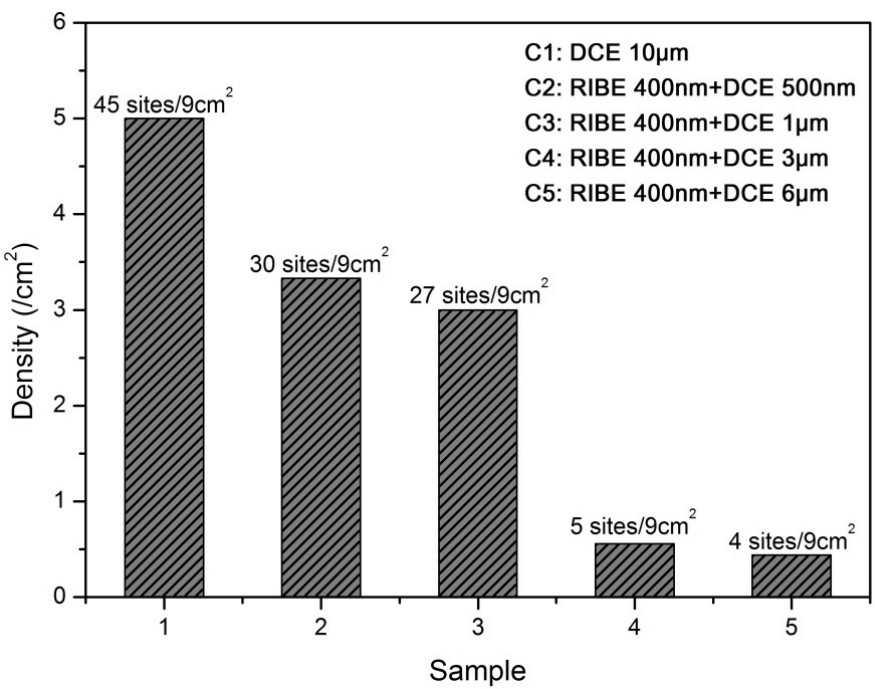

**Figure 9.** Laser damage initiation density at uniform fluence (11 J/cm$^2$@351 nm, 5 ns) for various fused silica-combined-etching processes (including a single DCE sample) is described in Table 1.

## 4. Conclusions

High-peak-power lasers require SSD removal or minimization to improve the laser-induced damage resistance of optics. This methodology can serve as a basis for estimating material removal during the combined etching of fused silica. Minimizing or eliminating SSD can lead to the manufacturing of optics that are both more economical and perform better. Thus, RIBE provides a means of characterizing subsurface damage distribution and the significantly greater precise and traceless elimination of SSD that can improve the damage resistance of fused silica optics.

**Author Contributions:** Conceptualization, Q.L.; Software, Y.Z. and Z.S.; Validation, T.S.; Formal analysis, Q.L., Y.Z., T.S., J.H. and X.Y.; Investigation, Y.Z. and T.S.; Data curation, Z.S., J.H. and L.Y.; Writing—original draft, Q.L.; Writing—review & editing, Q.L. and Y.Z.; Visualization, J.H. and W.Z.; Supervision, X.Y., L.Y. and W.Z.; Project administration, L.Y. and W.Z.; Funding acquisition, Q.L. and Z.S. All authors have read and agreed to the published version of the manuscript.

**Funding:** National Natural Science Foundation of China (NSFC) (62175222, 61705206, 62005258, and 61805221). Laser Fusion Research Center Funds for Young Talents (RCFPD3-2019-2). The Open Project Program of Key Laboratory for Cross-Scale Micro and Nano Manufacturing, Ministry of Education, Changchun University of Science and Technology (CMNM-KF 202110).

**Institutional Review Board Statement:** Not applicable.

**Informed Consent Statement:** Not applicable.

**Data Availability Statement:** The data presented in this study are not available due to privacy.

**Conflicts of Interest:** The authors declare no conflict of interest.

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
