# Peer review of "Rapid Detection and Elimination of Subsurface Mechanical Damage for Improving Laser-Induced Damage Performance of Fused Silica"

_coatings, doi:10.3390/coatings14040466_

Round 1

Reviewer 1 Report

Comments and Suggestions for Authors

The paper represents the results of an extensive study of elimination of subsurface mechanical damage in fused silica using a combination of reactive ion-beam- and chemical etching. In general, the paper is well written and the results are supported by the data; these results should represent an interest to laser optics community. In the Reviewer's opinion, the paper can be published in Coatings, provided some minor issues are addressed:

1) Text in line 105 refers to Figure 1(b). There is no Figure 1(b) in the paper.

2) Lines 137-138: 'with etching rates ranging from 0 nm to 1000 nm'. Rate cannot be measured in nanometers. The authors probably meant 'depths'.

3) 'DCE' is explained only in line 159, while used much earlier in the text. Also, in general, the abstract should be free from abbreviations, or they should be explained on the spot.

4) How the results shown in Figure 6 were obtained? Probably, using atomic  force microscopy? Some comments are needed in the text on that.

5) Lines 239-240. The readers are unlikely aware of the performance or technicaL details of 'UV laser damage platform of Department 7', please give some details on this experiment.

3) 

Comments on the Quality of English Language

In general, English language is fine. However, the expression "etch removal" causes some concern. The Reviewer understands that It is either 'etching' or 'material removal'.

Author Response

Dear editor and reviewer:

We are grateful to the editor and reviewers for their constructive comments and suggestions on the revision of the manuscript. We have made all the necessary changes again as suggested by the editor and reviewers. All the revisions in the manuscript and the supporting information have been highlighted in underline.

Reviewer #1:

Comment 1: Text in line 105 refers to Figure 1(b). There is no Figure 1(b) in the paper.Respond: Thank you very much for your comment.There was an error in our writing, which has now been corrected in the article, as follows:

We provide a schematic diagram of the ion beam etching principle in Figure 1.

Comment 2: Lines 137-138: 'with etching rates ranging from 0 nm to 1000 nm'. Rate cannot be measured in nanometers. The authors probably meant 'depths'.

Respond: Thank you for your advice. here was an error in our writing, which has now been corrected in the article, as follows:

We create a nanostructure using Reactive Ion Beam Etching (RIBE) with etching depths ranging from 0 nm to 1000 nm.

Comment 3: 'DCE' is explained only in line 159, while used much earlier in the text. Also, in general, the abstract should be free from abbreviations, or they should be explained on the spot.

Respond: Thank you very much for your comment. We have provided an explanatory note in the summary and have also corrected it in line 159.

To treat the surface of polished fused silica, a combination of RIE/RIBE and deep controlled etch (DCE) techniques are typically employed.

Our experimental design included five different RIBE depths (200 nm, 400 nm, 600 nm, 800 nm, and 1 μm) that corresponded to a 3-μm DCE.

Comment 4: How the results shown in Figure 6 were obtained? Probably, using atomic  force microscopy? Some comments are needed in the text on that.

Respond: Thank you very much for your comment. We have added to the test method and principle of Fig. 6 in the article and amended, as follows:

Figure 5 displays the optical micrographs of the six samples which underwent combined etching with different RIBE depths or treatment with 10 µm DCE. Figure 6 shows the surface morphology of the etched samples, which were exam-ined using a white light interferometer. The test process is to use the white light interference prin-ciple, record the position information in a certain area of the sample surface, and count, and get the surface roughness value of the optical element.

Comment 5: Lines 239-240. The readers are unlikely aware of the performance or technicaL details of 'UV laser damage platform of Department 7', please give some details on this experiment.

Respond: Thank you very much for your comment. Our article adds to the parameters related to damage testing as follows:

A deeper DCE treatment depth of 6 μm was added to observe its effect on the compo-nent damage threshold. sing the UV laser damage platform of Department 7(The wavelength, pulse width, and laser repetition rate were 355 nm, 9.3 ns (FWHM), and 1 Hz, respectively), the experimental study was conducted using the "raster-scan" damage testing strategy.

Thank you for your attention and patience, we hope the editors will take a second look at the work we're doing.

Yours sincerely.

Yubin Zhang

Reviewer 2 Report

Comments and Suggestions for Authors

The overall quality of the article is okay. I have a few comments or suggestions to make it better. I have added the suggestions below;

1.The roughness in figure 6 is required to be analyzed with some kind of statistical analysis to make it more relevant to the result

2.Figure 5 has only scale bar in the first image. Is it the same for the all-other images (b-f)? The author should mention it in the caption or add it in the figure itself

3.What are the procedures for determining the most appropriate depths of Reactive Ion Etching (RIE)/Reactive Ion Beam Etching (RIBE) and Dynamic Chemical Etching (DCE) to get rid of Subsurface Damage (SSD) on fused silica in order to maximize laser-induced damage threshold?

4.How effective is it to combine Reactive Ion Etching (RIE)/Reactive Ion Beam Etching (RIBE) with Dielectric Charged Etching (DCE) to remove Silicon Surface Damage (SSD) without creating new flaws, like redeposited silica compounds, that can hurt the performance of optical components?

5.How do variations in the etching depths of RIE/RIBE and DCE affect the laser damage resistance of fused silica, and what is the optimal matching relationship between these etching depths to significantly enhance the damage threshold?

Author Response

Dear editor and reviewer:

We are grateful to the editor and reviewers for their constructive comments and suggestions on the revision of the manuscript. We have made all the necessary changes again as suggested by the editor and reviewers. All the revisions in the manuscript and the supporting information have been highlighted in underline.

Reviewer #2:

Comment 1: The roughness in figure 6 is required to be analyzed with some kind of statistical analysis to make it more relevant to the result.Respond: Thank you for your advice. In performing the test in Figure 6, we selected a white light interferometer for the test. During the test, we recorded the position information of 307,200 points in a certain area of the sample surface and counted these points by software processing to get the surface roughness value of the optical element. Therefore, the data given in Figure 6 are already counted. We have supplemented the test technique of Fig. 6 in our article as follows:Figure 6 shows the surface morphology of the etched samples, which were exam-ined using a white light interferometer. The test process is to use the white light interference principle, record the position information in a certain area of the sample surface, and count, and get the surface roughness value of the optical element.

Comment 2:

.Figure 5 has only scale bar in the first image. Is it the same for the all-other images (b-f)? The author should mention it in the caption or add it in the figure itself.

Respond: Thank you for your advice. The scales are all consistent in Figure 5, and we have added a note in the caption as follows:

Figure 5. Optical micrographs of samples being single DCE treated with 3 μm and 10 μm material removal (a-b) and combined-etched samples being RIBE pretreated with 200 nm, 400 nm, 600 nm, and 800 nm material removal (c-f). The scales in (b-f) are all consistent with (a).

Comment 3: What are the procedures for determining the most appropriate depths of Reactive Ion Etching (RIE)/Reactive Ion Beam Etching (RIBE) and Dynamic Chemical Etching (DCE) to get rid of Subsurface Damage (SSD) on fused silica in order to maximize laser-induced damage threshold?

Respond: Thank you very much for your comment. The polished optical element consists of a damage precursor, a hydrolyzed layer, a subsurface defect layer, a structural defect layer and a substrate material from top to bottom. Typically, during the etching process, the damage precursor and hydrolysis layer are removed empirically by RIE/RIBE etching to expose the subsurface defects of the optical element. Sub-surface defects are then removed by etching using DCE technology. While the etching step is being performed, samples from each stage are retained for damage testing to increase the laser-induced damage threshold of the optics.

Comment 4: How effective is it to combine Reactive Ion Etching (RIE)/Reactive Ion Beam Etching (RIBE) with Dielectric Charged Etching (DCE) to remove Silicon Surface Damage (SSD) without creating new flaws, like redeposited silica compounds, that can hurt the performance of optical components?

Respond: Thank you very much for your comment. After etching the optical element by RIE/RIBE, the transmittance of the optical element is increased due to the formation of a sub-wavelength structure on the surface. After DCE treatment, the roughness of the optical element is increased due to the removal of sub-surface defects.

Comment 5: .How do variations in the etching depths of RIE/RIBE and DCE affect the laser damage resistance of fused silica, and what is the optimal matching relationship between these etching depths to significantly enhance the damage threshold?

Respond: Thank you very much for your comment. The purpose of RIE/RIBE etching is to remove surface defects and hydrolyzed layers generated by polishing of the optical element, ultimately exposing subsurface defects (e.g., metallic impurity elements, etc.) The purpose of DCE is to remove subsurface defects by chemical reaction. Therefore, the optimal value of RIE/RIBE etching depth is just enough to expose the subsurface defects of the optical element, and the optimal depth of DCE etching is the thickness of the subsurface defect layer. If the RIE/RIBE as well as DCE etching depths are not sufficient, this will result in the optics still having polishing defects and a lower threshold. If the etching depth is too much, the substrate material is affected, which also leads to a lower threshold.

Thank you for your attention and patience, we hope the editors will take a second look at the work we're doing.

Yours sincerely.

Yubin Zhang